# Nonlinear synchronization through vector subharmonic entrainment
Dmitrii Stoliarov [1,3] ✉, Sergey Sergeyev[1,3], Hani Kbashi[1], Fan Wu[2], Qianqian Huang[2] & Chengbo Mou [2,3] ✉

Synchronization is a universal phenomenon underpinning various natural processes and finds direct application in control engineering and photonics. Among several synchronization mechanisms, subharmonic entrainment (SHE) is a nonlinear synchronization phenomenon where an oscillator locks to an external drive with a fraction of the oscillator's frequency. While its mechanism is well understood for scalar couplings and finds application in the stabilization of ultrafast laser pulses, the potential of SHE with vectorial coupling is still unexplored. In this work, we demonstrate vector SHE (VSHE) using a passively mode-locked fiber laser as a testbed. We unveil the mechanism of vector SHE, in which weak external signals can entrain internal laser dynamics through vector coupling. Vector SHE presents in the form of synchronization between the subharmonic of mode-locking-driven oscillations and continuous wave (CW) signal with an evolving state of polarization. This CW signal, driven by the internal dynamics of the injected signal, causes VSHE with the frequencies' ratios of multiples of ten, resulting in a partially mode locking regime operation. Our findings offer new control techniques over mode-locking and additional dimension such as polarization states.

Synchronization phenomena, such as those described by coupled oscillator models—where multiple interacting systems spontaneously coordinate their dynamics—are fundamental in various natural and engineered networks[1–9]. Typically, the research agenda focuses on revealing mechanisms of how the interactions between individual oscillators driven by localized external perturbations caused by the injected signal result in synchronization (injection locking), e.g., the capability of coupled oscillators to synchronize at a common frequency[1–9]. For example, in the field of photonics, injection locking of lasers was initially motivated by the ability to stabilize high-power continuous wave lasers by using low-power master lasers[10,11]. Subsequent studies have investigated various injection locking techniques for stabilizing pulsed lasers, such as mode-locked lasers' pulse trains[12,13], multi-wavelength solitons[14], vector soliton rain dynamics control[15], bound state solitons[16,17], and both tunable[18–20] and low-noise harmonic mode-locked pulses[21,22]. In particular, various synchronization scenarios driven by internal dynamics have been explored. These include frequency locking with free-running phase[23], subharmonic entrainment of pulsating single solitons in an ultrafast laser leading to the breathing dynamics[24,25], synchronization of polarization modes with zero-log, phase difference entrainment and desynchronization[26–34].

SHE is a form of synchronization in which phase locking occurs between coupled oscillators or between an oscillator and an injected signal[9,24,35]. In photonics, SHE provides an effective method for phase and frequency locking, eliminating the necessity for strong actuation or complicated feedback systems. In ultrafast and mode-locked lasers, SHE can suppress timing jitter, stabilize pulse trains, and tame unwanted dynamical states by synchronizing internal oscillations to an external periodic drive, which is attractive for robust operation in demanding applications. Theoretically, SHE arises when an oscillator with a frequency $\omega$ synchronizes to a lower frequency $\omega_0$ typically satisfying a rational ratio $\omega_0/\omega < 1$. These synchronization regimes, known as Arnold tongues, allow phase locking even under small detuning, provided the coupling strength is sufficient. In particular, the synchronization region for SHE with $\omega_0/\omega = 0.5$ can be relatively broad[9]. However, most demonstrations and models treat the SHE as essentially scalar, focusing on frequency or amplitude/timing locking and simple fractional ratios[24,35], even though many laser systems are inherently vectorial because of polarization dynamics and the presence of birefringent, anisotropic, or polarization-dependent components[36–40].

Studying the vector counterpart, such as VSHE, adds an alternative control knob: rather than acting only on intensity or phase, weak external signals can couple through polarization and steer the internal dynamics via an evolving state of polarization. This opens opportunities for polarization-selective stabilization, controlled switching between dynamical regimes, access to partially mode-locked states and pulse properties by targeting vector degrees of freedom that are otherwise difficult to control.

[1]Aston Institute of Photonics Technologies, Aston University, Birmingham, UK. [2]Key Laboratory of Specialty Fiber Optics and Optical Access Networks, Joint International Research Laboratory of Specialty Fiber Optics and Advanced Communication, Shanghai University, Shanghai, China. [3]These authors contributed equally: Dmitrii Stoliarov, Sergey Sergeyev, Chengbo Mou. ✉e-mail: d.stoliarov@aston.ac.uk; mouc1@shu.edu.cn

As illustrative example, we consider the spring-coupled simple pendulum system in Fig. 1a. This system provides a classical analog to the dynamics of orthogonally polarized modes in mode-locked lasers, where each pendulum represents an oscillator associated with one of the state of polarization (SOP)[8,9,26,30]. The synchronization regimes of the coupled oscillators can be explained by the general Adler equation, describing the evolution of the phase difference $\Delta\phi$ between two coupled oscillators[8,9,26,28,30]:

$$\frac{d\Delta\varphi}{dt} = \Delta\Omega + K \cdot sin(\Delta\varphi) \qquad (1)$$

where $\Delta\Omega$ is the frequency difference, and K is the coupling coefficient. For polarization dynamics of the mode-locked lasers, the frequency difference is the function of the linear and circular birefringence, and the coupling coefficient is a function of the output powers of the orthogonal SOPs[26,28,30]. As follows from the Eq. (1), the phase-locked synchronization ($d\Delta\varphi/dt = 0$) exists when $|\Delta\Omega| < |K|$ that corresponds to the continuous-wave (CW) mode-locking[26,28,30]. Unlike the phase-locking,

condition $|\Delta\Omega| > |K|$ means the phase entrainment, e. g. phase difference oscillations[8,26,28,30]. In contrast to the classical Adler equation with constant coefficients, the generalized Adler equation derived for the polarization laser dynamics comprises the time-dependent coefficients[26,28,30]. As a result, condition of the phase entrainment has to be generalized[28]. Though differences between mechanical model and vector mode-locked laser, phase-difference dynamics in mode-locked fiber lasers can be treated in terms of coupling-induced transitions[26,28,30].

In this work, we report the experimental and theoretical demonstration of vector subharmonic entrainment (VSHE) in an ultrafast laser. We experimentally observe synchronization between a low-frequency, injected, polarization-modulated signal and internal polarization oscillations in an ultrafast laser. This phenomenon results in a double-timescale pulsation regime, also known as Q-switched mode-locked (QSML), characterized by a train of ultrashort pulses modulated by a slowly varying envelope. The observed synchronization scenarios were mapped by the phase difference dynamics and were adjusted by varying the injected signal modulation frequency, amplitude, and the state of polarization synchronization regime.

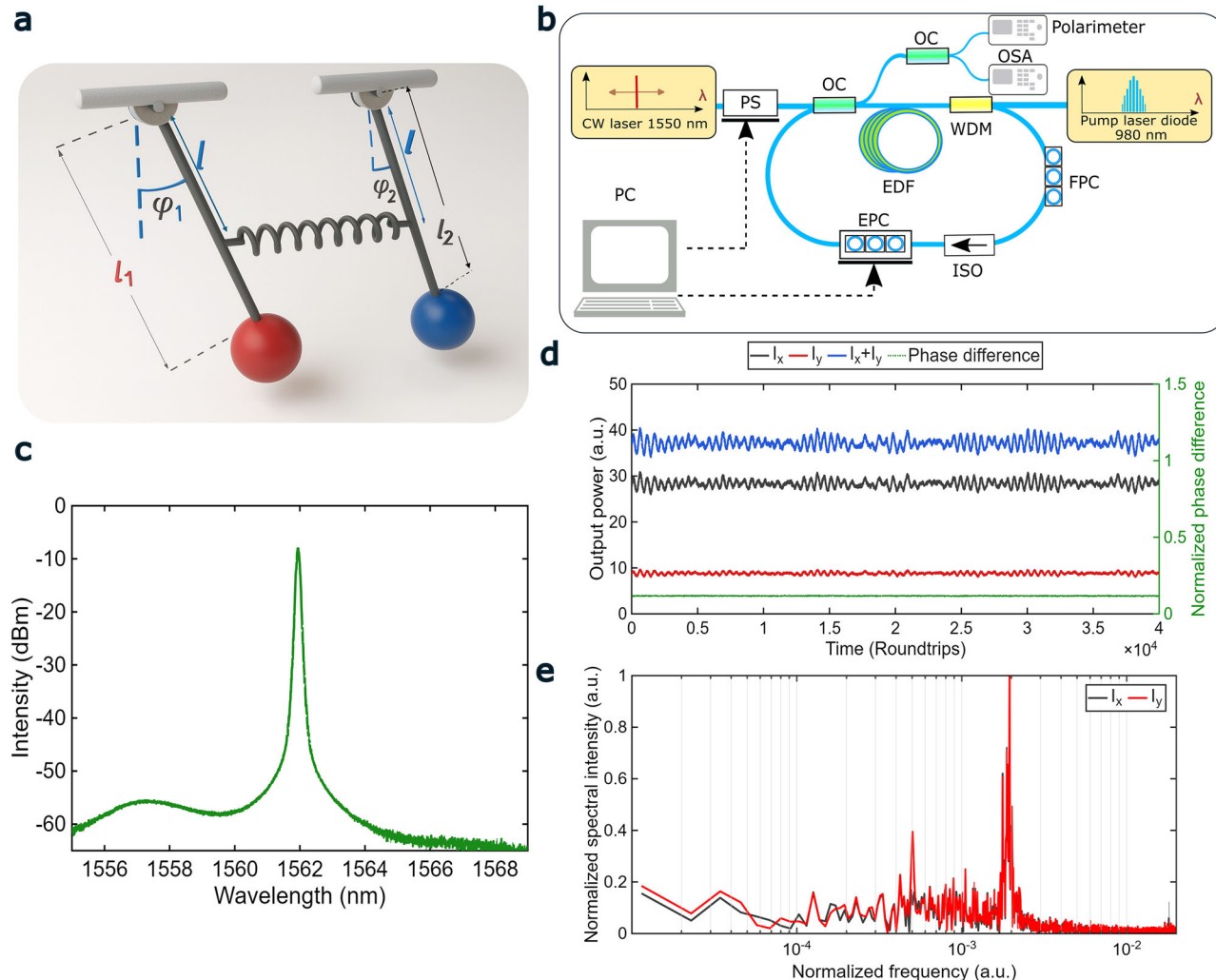

**Fig. 1 | Principle and schematic of subharmonic entrainment induced nonlinear synchronization. a** Simple spring-coupled oscillator model: two pendulums of different mass and unequal lengths $l_1$ and $l_2$ are coherently coupled via a spring of finite stiffness. The angular displacements $\varphi_1$ and $\varphi_2$ describe the phase evolution of each oscillator, which possess distinct natural frequencies; **b** Schematic setup of the fiber laser. EDF erbium-doped fiber, 980 nm pump laser diode; 1550 nm CW laser continuous wave laser, FPC manual fiber polarization controller, EPC electronically driven polarization controller, ISO polarization-sensitive optical isolator, WDM wavelength-division multiplexer, PS, polarization scrambler, OC 70/30 optical coupler, OSA optical spectrum analyser; **c** Optical spectrum registered at the OSA of the CW regime of Er-doped fiber laser; **d** Polarization dynamics of Er-doped fiber laser registered at the fast polarimeter in terms of the output powers oscillations of the individual polarization components $I_x$ (black) and $I_y$ (red), total power $I = I_x + I_y$ (blue), and the phase difference $\Delta\varphi$ between orthogonal x- and y-SOPs normalized to $\pi$ (green); **e** Fast Fourier transform spectrum of the x (black) and y (red) SOPs' power dynamics with frequencies normalized to the fundamental frequency $f_0 = 16.67$ MHz. Parameters: pump power $I_p = 72$ mW, averaging time for the polarimeter trace $T_{pol} = 320$ ns.

## Results

### Experimental demonstration

To investigate VSHE, we built a nonlinear polarization rotation (NPR)-based mode-locked fiber ring laser operating under the anomalous dispersion regime. A schematic representation of the mode-locked fiber laser configuration is shown in Fig. 1b, the detailed description of the setup is provided in the Methods section.

We seeded an optical signal with power of 15 dBm and modulation frequency of 1.67 kHz through a vacant input port of a 30/70 output optical coupler. A fast polarimeter PM1000-XL-FA-N20 D (Novoptel) with a sampling frequency of 100 MS/s was used to observe the evolution of the Stokes parameters $S_1$, $S_2$, and $S_3$, the power for the orthogonal x- and y-polarization components ($I_x$, $I_y$), the total power $I$, and the phase difference $\Delta\varphi$ was recalculated from the measured Stokes parameters as follows:

$$S_0 \equiv I = I_x + I_y, S_1 = I_x - I_y,$$
$$S_2 = 2\sqrt{I_x I_y}\cos(\Delta\varphi), \qquad (2)$$
$$S_3 = 2\sqrt{I_x I_y}\sin(\Delta\varphi).$$

The polarization resolved results of our experimental study are shown in Fig. 1. During the experiments, the pump power was maintained at a constant level of $I_p = 72$ mW, with only the polarization controllers being adjusted. In the absence of the injected signal, the electronically driven polarization controller (EPC) was tuned to establish the CW regime, the optical spectrum registered at the OSA (optical spectrum analyser) is shown in Fig. 1c. As illustrated, the optical spectrum exhibits a localized, narrowband spike-like structure centered around 1562 nm. The polarization resolved temporal behaviors of pulses, shown in Fig. 1d, appear as a close to CW low-amplitude oscillations in the output powers of the orthogonal SOPs, $I_x$ and $I_y$, the total power $I = I_x + I_y$, and the phase difference $\Delta\varphi$. A fast Fourier transform (FFT) reveals that the oscillation frequency is $f_{CW} = 2 \times 10^{-3} \cdot f_0$ for the x- and y-SOPs, as shown in Fig. 1e. Given that the phase difference is constant, we have a typical case of a phase-locked regime[9].

The developed test bed is well suited for studying internal nonlinear polarization dynamics within a fiber laser cavity. In this setup, a QSML regime is observed, which is driven by an external CW signal. This regime exhibits two-dimensional high-order partial synchronization across both temporal and polarization domains. Such behavior provides a robust experimental platform for investigating vector synchronization in nonlinear optical systems.

By further adjusting the polarization controller, we were able to induce QSML dynamics. As seen in Fig. 2b and Fig. 2d, the optical spectrum indicates the presence of a CW component with the peak wavelength at 1562 nm and frequency $f_{CW} = 2 \times 10^{-3} \cdot f_0$, similar to the previous case shown in Fig. 1c, e.

The structure of a single fringe in the time domain and the partial mode-locking regime is shown in Fig. 2a and its inset. Slow QSML polarization dynamics are illustrated in Fig. 2c, where the dynamics exhibit double-scale oscillations in the output powers of the orthogonal polarization components: $I_x$ (black), $I_y$ (red), and total power $I = I_x + I_y$ (blue), along with the normalized phase difference $\Delta\varphi$ showing phase difference oscillations with slips of approximately $\pi$ radians (green).

A fast Fourier transform reveals distinct spectral components in the power dynamics of the x- and y-SOPs: low-frequency oscillations at $f_{LQ1} = 4 \times 10^{-5} \cdot f_0$ and $f_{LQ2} = 8 \times 10^{-5} \cdot f_0$, and a high-frequency component at $f_{HQ} = 2 \times 10^{-3} \cdot f_0$ with multiple sidebands. The presence of sidebands in the high-frequency oscillations is the result of overlapping with the low-frequency components, thereby enabling partial synchronization through the SHE mechanism.

To further investigate the VSHE dynamics driven by a low-frequency CW component in the cavity, an external CW laser with a deterministic state of polarization was used to inject a modulated low-frequency signal at 1562 nm. The central wavelength of the injected signal was detuned to

match the central wavelength of self-started internal CW generation within the laser cavity, consistent with the configuration shown in Fig. 1c and Fig. 2b. The results, shown in Fig. 3 a–c, align closely with those in Fig. 2 in terms of optical spectrum, polarization-resolved temporal dynamics, and the Fourier spectral components. Similarly to the dynamics in Fig. 2d, the slow QSML dynamics again reveal two characteristic frequency scales: $f_{LQ1} = 10^{-4} \cdot f_0$, $f_{HQ1} = 10^{-3} \cdot f_0$, and $f_{HQ2} = 2 \times 10^{-3} \cdot f_0$.

### Theoretical Analysis

To explain the mechanism of the vector SHE, we explore a vector model of an NPR-based Er-doped mode-locked fiber laser with an injected signal (see Method section). An approach which was previously used for theoretical characterization of NPR-based mode-locking is based on the introduction of function q which is related to amplitudes of the polarization components $u$ and $v$ as follows[36]:

$$u = q \cdot \cos(\alpha_p), v = q \cdot \sin(\alpha_p). \qquad (3)$$

Here, $\alpha_p$ is the angle between the polarization plane of the polarizer and the slow axis[36]. The complex amplitudes of the orthogonally polarized SOPs, $u$ and $v$, can also be expressed in terms of the corresponding norms of the complex numbers $|u|$, $|v|$ and phases $\varphi_x$, $\varphi_y$ as follows:

$$u = |u| \cdot \exp(i \cdot \varphi_x), v = |v| \cdot \exp(i \cdot \varphi_y). \qquad (4)$$

As follows from Eqs. (3) and (4), the phase difference is $\Delta\varphi = \varphi_y - \varphi_x \equiv 0$, and the polarization angle $\alpha_p$ satisfies $\tan(\alpha_p) = |v|/|u|$. However, as follows from Figs. 2c and 3b, the phase difference $\Delta\varphi$ is oscillating, which makes approach developed in ref. 36 not acceptable for the modeling of the polarization dynamics observed in our experiments.

To overcome the drawbacks of previously used model[36] and so reveal mechanism of the mode-locking dynamics driven by the SHE between low-frequency oscillations, we updated a vector model of an Er-doped mode-locked fiber laser recently developed by Sergeyev and co-workers (details are found in refs. 26–31). The model describes the evolution of the SOP of the lasing averaged over the pulse width driven by in-cavity CW signal with periodically evolving orthogonal states of polarization:

$$E_x = a \cdot \cos(\Omega t + \phi_0), E_y = a \cdot \sin(\Omega t + \phi_0) \cdot \exp(i \cdot \Delta\Phi) \qquad (5)$$

Here, $a$ is the amplitude of the injected optical signal, $\Omega$ is the frequency of oscillations normalized to the fundamental frequency $f_0$, $\varphi_0$ is the initial phase, and $\Delta\Phi$ is the phase difference between the orthogonal SOPs. As follows from Eq. (5), the injected signal is CW, i.e., the total power is given by $I = I_x + I_y = |E_x|^2 + |E_y|^2 = a^2 = \text{const}$.

To gain an in-depth understanding, we further carried out a linear stability analysis (see Supplementary Information Note 2) to aid the identification of NPR-based oscillations with a frequency normalized to the fundamental frequency $f_0$ of approximately as $f_{NPR}/f_0 = 4 \times 10^{-3}$, which closely matches the experimentally observed results (Fig. 1c, Fig. 2c, and Fig. 3c).

By adding a CW signal into the cavity with parameters $a = 0.19$, $\Omega = 2\pi \cdot 0.00005$, $\varphi_0 = \pi/4$, and $\Delta\Psi = -\pi/4$, the QSML dynamics emerge, as shown in Fig. 4a, b.

The same as for typical injection locking, detuning, e.g., frequency change to $\Omega = 2\pi \cdot 0.0001$, results in increased frequency of the pulse bunching as shown in Fig. 4c, d.

As follows from Fig. 4a, b, the QSML dynamics of the output powers for polarization modes and total is similar to the experimentally observed (Fig.2c and Fig.3b) in the context of the shape of pulses, the two-scale oscillations in the range of the normalized frequencies with $10^{-4}$ and $10^{-3}$, and the number of sidebands (Fig.4b), and the phase difference slips by about $\pi$ radians. With increased amplitude of the injected signal to $a = 0.5$,

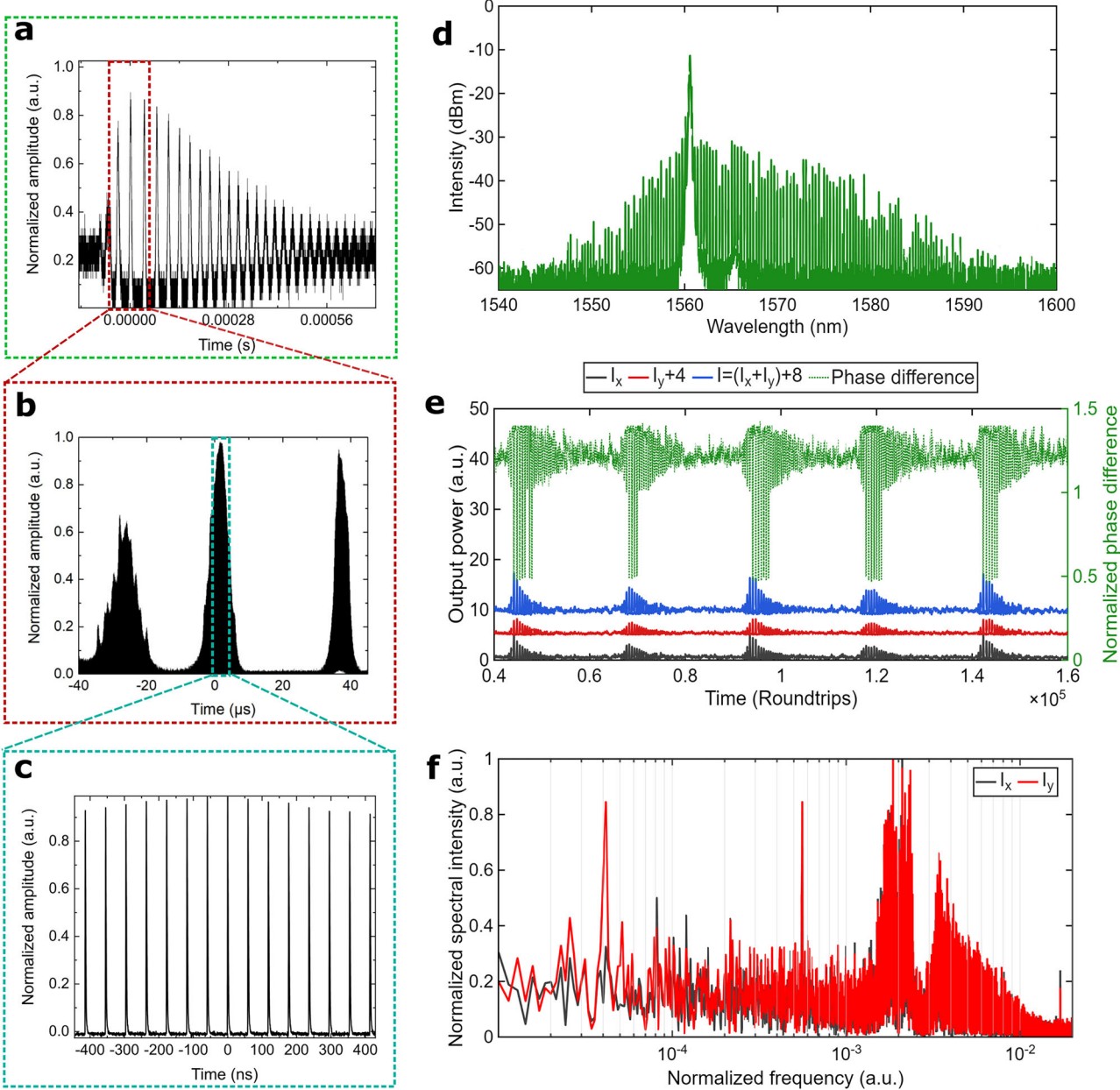

**Fig. 2 | Partially mode-locking regime of ultrafast laser. a–c** Oscillogram at different timescales: **a** second, **b** microsecond, **c** nanosecond; **d** Optical spectrum registered at the OSA in the partially mode-locked regime. **e** Polarization dynamics of the individual polarization components $I_x$ (black) and $I_y$ (red), total power I =

$I_x + I_y$ (blue), and the phase difference $\Delta\varphi$ between orthogonal x- and y-SOPs, normalized to π (green, Secondary Y-axis), I and $I_y$ are shifted for better representation; **f** Fast Fourier transform spectrum of the x (black) and y (red) SOPs' power dynamics. The averaging time for the polarimeter trace is $T_{pol} = 1.28$ μs.

the QSML dynamics transform into a spiking regime with the normalized frequency $f_1 = 10^{-4}$, as shown in Fig. 5a, b.

To specify the effect of the rotating state of polarization, we considered an additional case with $\Delta\Phi = -\pi/2$ and $a = 0.19$. The breathing dynamics for this case are shown in Fig. 5c, d. The spiking dynamics is quite similar to the dynamics shown in Fig. 5a, b. The main differences are in increased pulse bunches oscillation frequency and reduced amplitude of the injected signal.

## Discussion

We revealed experimentally and theoretically that our vector type of subharmonic entrainment causes double-scale mode-locking dynamics. Unlike the previous study of SHE related to the synchronization of the subharmonic of the fundamental frequency and NPR-driven oscillations[24,25], it was shown both experimentally and theoretically that vector SHE takes place through overlapping and synchronization of the sidebands of the NPR

with low-frequency oscillations having evolving state of polarization. The FFT of the dynamic waveforms demonstrate the increased number of the sidebands around the high-frequency component with increased amplitude and varying dynamic SOP of the injection signal, which indicates enhanced coupling between the high- and low-frequency oscillations. The observed SHE represents a distinct synchronization phenomenon where, as shown in refs. 9,25, the synchronization behavior depends on both the injected signal amplitude (or coupling coefficient) and the detuning between the injected modulated frequency and the internal oscillation frequency of the system. According to the synchronization theory for subharmonic entrainment in our case, the ratio of frequencies can slightly deviate from an integer, which results in the synchronization in the form of phase difference entrainment, e.g., the phase difference oscillations shown in refs. 9,25,36 and Figs. 2–4. With an increased amplitude of the injected signal (see Fig. 4), in line with the synchronization theory[9,25,36], we observe phase- and frequency locking

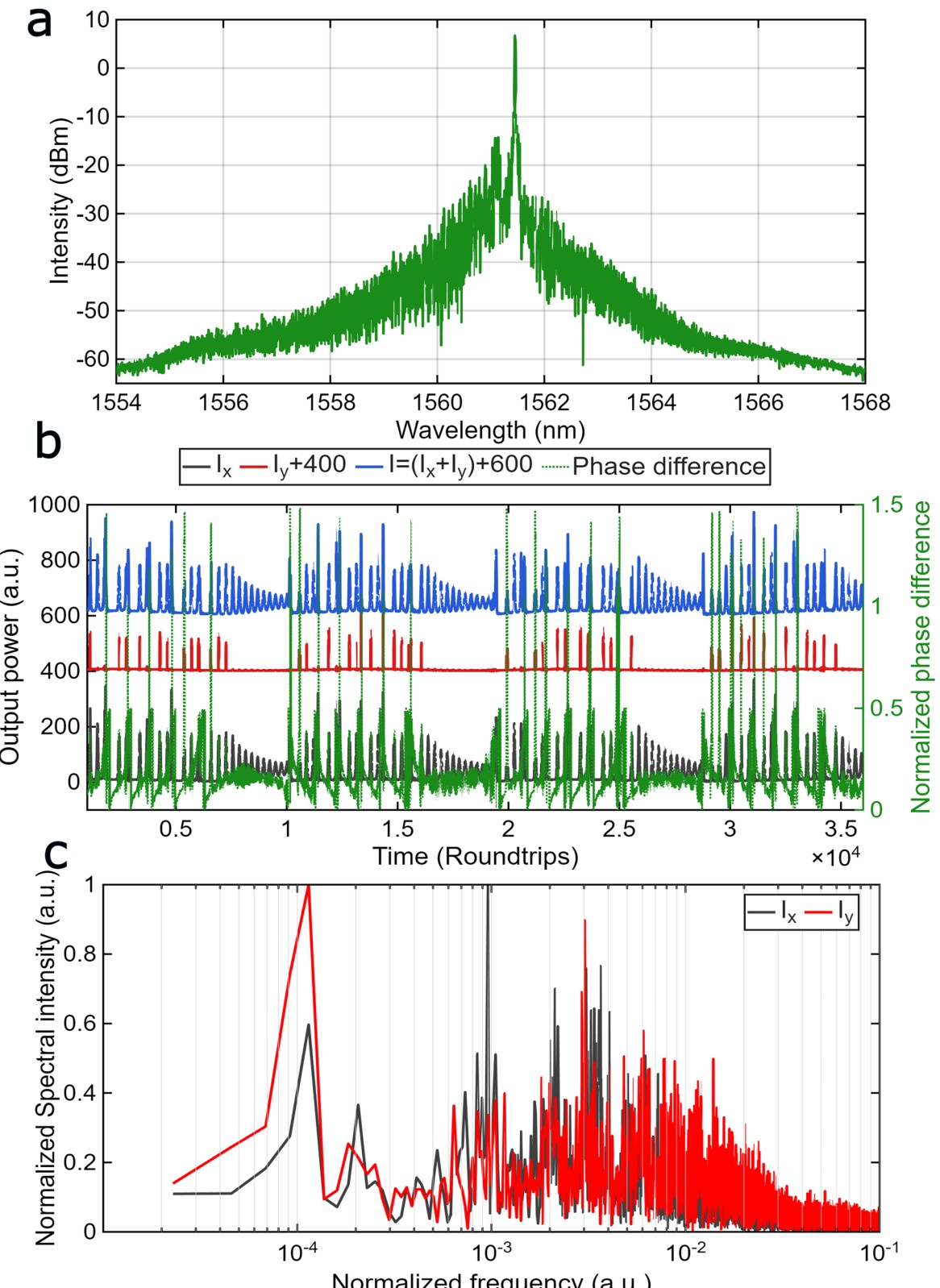

**Fig. 3 | Vector subharmonic entrainment synchronized regime of ultrafast laser.**
**a** Measured at the OSA optical spectrum; **b** Polarization dynamics of the individual polarization components $I_x$ (black) and $I_y$ (red), total power $I = I_x + I_y$ (blue), and the phase difference $\Delta\varphi$ between orthogonal x- and y-SOPs, normalized to $\pi$ (green, Secondary Y-axis), I and Iy are shifted for better representation; **c** Fast Fourier transform spectrum of the x (black) and y (red) SOPs' power dynamics. The averaging time for the polarimeter trace $T_{pol} = 320\ ns$.

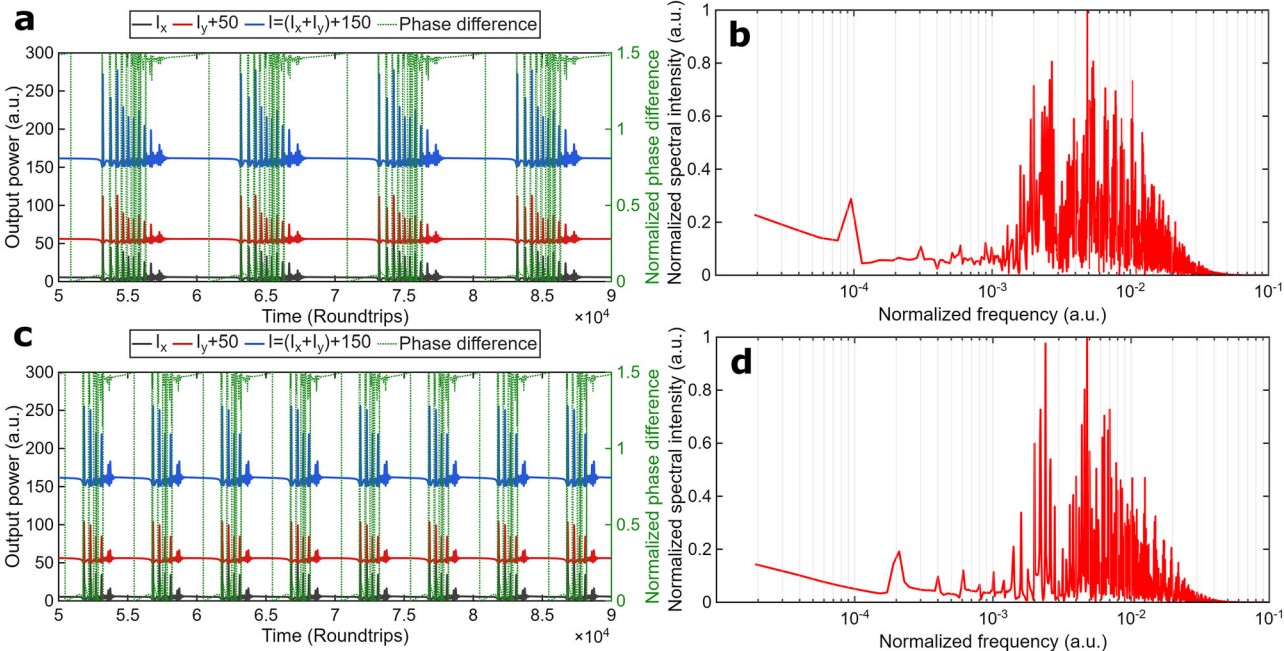

**Fig. 4 | Theoretical results on vector SHE dynamics.** Parameters: injected signal with the phase difference of $\Delta\Phi = -\pi/4$ and amplitude $a = 0.19$, injection frequency $\Omega = 2\pi \cdot 0.00005$ (**a,b**) and $\Omega = 2\pi \cdot 0.0001$ (**c,d**): **a, c** Polarization dynamics of the individual polarization components $I_x$ (black) and $I_y$ (red), total power $I = I_x + I_y$ (blue), and the phase difference $\Delta\varphi$ between orthogonal x- and y-SOPs, normalized to $\pi$ (green, Secondary Y-axis), I and $I_y$ are shifted for better representation; **b, d** Fast Fourier transform spectrum of the y-SOPs' power dynamics with frequencies normalized to the fundamental frequency.

towards high-power oscillations. To obtain experimentally such dynamics, it is necessary to adjust the power and wavelength of the internal low-frequency oscillations, which is rather cumbersome for discussion here and will be published elsewhere. The distinct feature of vector SHE is the dependence of the synchronization regime on the polarization waveform of the injected signal, as shown in Figs. 4 and 5. The shape of the SOP of the injected signal with the rotating state of polarization enables control of the dwelling time near each orthogonal SOP, resulting in different types of switching between the orthogonal SOPs, taking the form of different QSML dynamics. This phenomenon is not only of fundamental interest for laser dynamics but also suggests avenues for photonic control[23]. By tuning the injection parameters-frequency, polarization modulation, and amplitude-one can selectively control envelope timing and polarization state. This could prove useful in ultrafast optical communication, pulse shaping, and metrology, where structured pulse sequences and polarization encoding are advantageous. Crucially, we demonstrate that subharmonic synchronization can occur even when the frequency ratio is not an exact integer.

In summary, vector subharmonic entrainment offers a lens through which to view and manipulate laser dynamics. The ability to externally control dual-frequency polarization-encoded pulse trains opens possibilities for advanced photonic systems and invites further exploration into the interplay of vector dynamics and synchronization. Our findings also connect to broader themes in nonlinear science, including synchronization of high-dimensional and vector systems. The analogy between polarization modes in lasers and multi-dimensional coupled oscillators offer a rich framework for exploring complex dynamics, with potential implications for understanding synchronization in biological, chemical, and engineered systems[1–9].

## Methods
### Experimental setup
The system utilizes a short piece of single-mode highly Erbium-doped optical fiber (EDF) with a peak core absorption of around 110 dB/m at 1530 nm, cutoff wavelength of 890 nm, the numerical aperture of 0.2. The EDF is pumped by a 976-nm pigtailed laser diode (pump laser diode)

through a 980/1550 nm wavelength division multiplexer (WDM). The EDF is subsequently spliced to the output port of a dual-stage polarization-sensitive optical isolator (polarization-sensitive ISO) with a center wavelength of 1550 nm and an extinction ratio of 28 dB. The 70/30 optical coupler (OC), featuring a coupling ratio centered at 1550 nm and an insertion loss around 1 dB, was employed to extract 30% of the laser power from the cavity. In addition, the configuration includes two fiber polarization controllers placed before and after the ISO. The first one is the in-line miniature manual in-line fiber polarization controller (FPC). The second one is the in-line EPC with a three-section external voltage control. The EPC software enables full state of polarization control through thermal technology, ensuring stable polarization scanning across the entire Poincaré sphere. It is important to note that, except for the ISO and EDF, the full cavity of the laser was constructed using standard fiber SMF-28. The total fiber laser cavity length is 12.3 m comprising 0.2 m EDF, 1.5 m PM-1550 fiber, and 10.5 m of SMF-28 fiber. This length also includes pigtail fibers from the EPC, isolator, and coupler. The anomalous group velocity dispersion (GVD) values of the SMF-28 and PM-1550 fibers at 1.55 μm were approximately 23 ps²/km and 22 ps²/km, respectively. The normal GVD of the Er-doped fiber at 1550 nm was estimated to be 15.3 ps²/km. As a result, the net dispersion of the cavity could be calculated as −0.27 ps², suggesting that the laser was operating at an anomalous dispersion regime. For optical signal injection, a Polarization Scrambler (PS) EPS1000 (Novoptel) and a narrow linewidth continuous wave laser (CoBrite DX1, IDPhotonics) are employed. The output light pulses were characterized and analyzed using various detection and measurement techniques, including a 50 GHz High-Speed Photodetector, a real-time oscilloscope with 6 GHz bandwidth, and an optical spectrum analyser AQ6317B (Yokogawa). All optical spectra were measured at the same wavelength resolution of 0.1 nm.

### Vector model of NPR mode-locked fiber laser with injected signal
The model includes a distributed form of Jones matrix comprising a combination of Jones matrices for two polarization controllers, (FPC and EPC), with a polarizer (ISO), where $\xi_{1(2)}$ is the angle of rotation of the vertical birefringent axis and $\varphi_{1(2)}$ is the phase shift between the wave components in

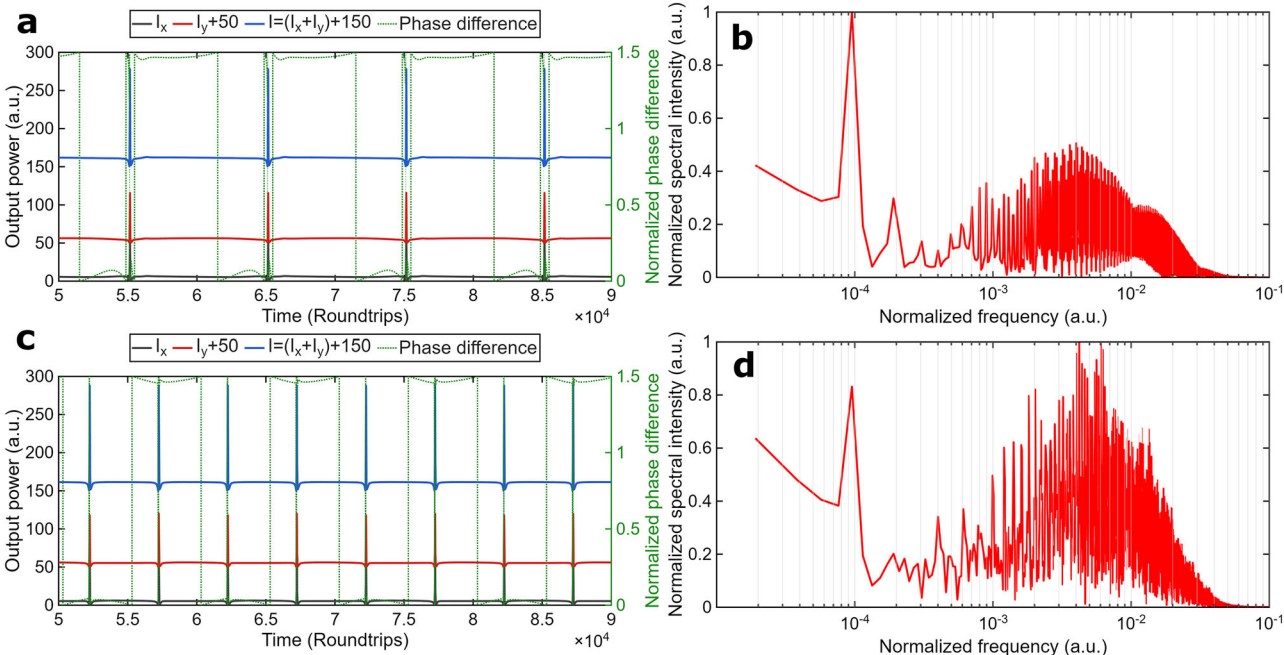

**Fig. 5 | Theoretical results on vector SHE dynamics.** Parameters: injected signal with the phase difference (**a,b**) $\Delta\Phi = -\pi/4$ and amplitude $a = 0.5$, (**c,d**) $\Delta\Phi = -\pi/2$ and amplitude $a = 0.19$: **a, c** Polarization dynamics of the individual polarization components $I_x$ (black) and $I_y$ (red), total power $I = I_x + I_y$ (blue), and the phase difference $\Delta\varphi$ between orthogonal x- and y-SOPs, normalized to $\pi$ (green, Secondary Y-axis), I and $I_y$ are shifted for better representation; **b, d** Fast Fourier transform spectrum of the y-SOPs' power dynamics with frequencies normalized to the fundamental frequency.

the two orthogonal birefringent axes. The angle between the axis of the polarizer and the $y$ axis is $\theta$.

Amplitudes of the cross-polarized states of polarization of the injected signal are $E_x = a \cdot \cos(\Omega t + \phi_0)$, $E_y = a \cdot \sin(\Omega t + \phi_0) \cdot \exp(i \cdot \Delta\Phi)$, ($a$ is the amplitude of the injected optical signal, $\Omega$ is the frequency of oscillations normalized to the fundamental frequency $f_0$, $\varphi_0$ is the initial phase, and $\Delta\Phi$ is the phase difference between the orthogonal SOPs); $u = |u| \cdot \exp(i \cdot \varphi_x)$, $v = |v| \cdot \exp(i \cdot \varphi_y)$, $\Delta\varphi = \varphi_y - \varphi_x$, $|u|^2 = I_x$, $|v|^2 = I_y$ are normalized to the saturation power $I_{ss}$ and $I_p$ is normalized to the saturation power $I_{ps}$.

With accounting for Eq. (5), the evolution equations for the complex amplitudes of the lasing field, averaged over the pulse width, and for the population inversion are as follows[26,28,30,31]:

$$\frac{du}{dt_s} = i\frac{\gamma}{2}\left(|u|^2 u + \frac{2}{3}|v|^2 u + \frac{1}{3}v^2 u^*\right) + \left(D_{xx} + A_{11}\right)u + \left(D_{xy} + A_{12}\right)v + E_x,$$

$$\frac{dv}{dt_s} = i\frac{\gamma}{2}\left(|v|^2 v + \frac{2}{3}|u|^2 v + \frac{1}{3}u^2 v^*\right) + \left(D_{xy} + A_{21}\right)u + \left(D_{yy} + A_{22}\right)v + E_y,$$

$$(6)$$

Here, $u$ and $v$ denote the complex amplitudes of the two orthogonally SOPs, as defined in Eq. (4). Their squared magnitudes correspond to the normalized lasing powers:

$|u|^2 = I_x$ and $|v|^2 = I_y$, $\gamma$ is normalized to the cavity length and the saturation power the Kerr constantt. The coefficients $D_{xx}$, $D_{xy}$, and $D_{yy}$, which describe the coupling and gain dynamics in the system, are introduced and explicitly defined in Eq. (7).

$$D_{xx} = \left(\frac{\alpha_1(1-i\Delta)}{1+\Delta^2}(f_1 + f_2) - \alpha_2\right),$$

$$D_{xy} = D_{yx} = \left(\frac{\alpha_1(1-i\Delta)}{1+\Delta^2}f_3\right),$$

$$D_{yy} = \left(\frac{\alpha_1(1-i\Delta)}{1+\Delta^2}(f_1 - f_2) - \alpha_2\right).$$

$$(7)$$

Here $\alpha_1$ is the total absorption of Erbium ions at the lasing wavelength, $\alpha_2$ represents the normalized losses and $\delta$ is the ellipticity of the pump wave, $f_1$,

$f_2$, and $f_3$ represent parameters associated with the angular distribution of excited ions, and their time evolution is governed by the system of differential equations provided in Eq. (8) where the variable $t_s$ denotes the slow time, normalized to the cavity round-trip time $\tau_r$.

$$\frac{df_1}{dt_s} = \varepsilon(b_1 - a_1 f_1 - a_2 f_2 - a_3 f_3),$$

$$\frac{df_2}{dt_s} = \varepsilon\left(b_2 - \frac{a_2}{2}f_1 - a_1 f_2\right),$$

$$\frac{df_3}{dt_s} = -\varepsilon\left(\frac{a_3}{2}f_1 + a_1 f_3\right).$$

$$(8)$$

Where, $\varepsilon = \tau_R/\tau_{Er}$ indicates the ratio of the cavity round-trip time $\tau_R$ to the lifetime of erbium ions in the first excited state $\tau_{Er}$. The coefficients $a$ and $b$ are defined in Eqs. (9) and (10).

$$a_1 = 1 + \frac{I_p}{2} + \frac{\chi(|u|^2+|v|^2)}{1+\Delta^2},$$

$$a_2 = \frac{\chi(|u|^2-|v|^2)}{1+\Delta^2} + \frac{I_p}{2}\frac{(1-\delta^2)}{1+\delta^2},$$

$$a_3 = \frac{\chi(u\cdot v^*+c.c.)}{1+\Delta^2}.$$

$$(9)$$

$$b_1 = \frac{(\chi-1)}{2}I_p - 1, \quad b_2 = \frac{I_p}{4}(\chi-1)\frac{(1-\delta^2)}{1+\delta^2}.$$

$$(10)$$

Also, $\chi = (\sigma_a + \sigma_e)/\sigma_a$, where $\sigma_a$ and $\sigma_e$ are absorption and emission cross-sections; $\Delta$ is the detuning of the lasing wavelength with respect to the maximum of the gain spectrum (normalized to the gain spectral width).

The transfer matrix for the combination of two polarization controllers, POC1 and POC2 (corresponding to Polarization Controllers PC and EPC in Fig. 1, respectively; see supplementary Information for a more detailed scheme Fig. S1), and a polarizer with polarization extension ratio of $p$ (POL, which in this case corresponds to the isolator in Fig. 1S) is

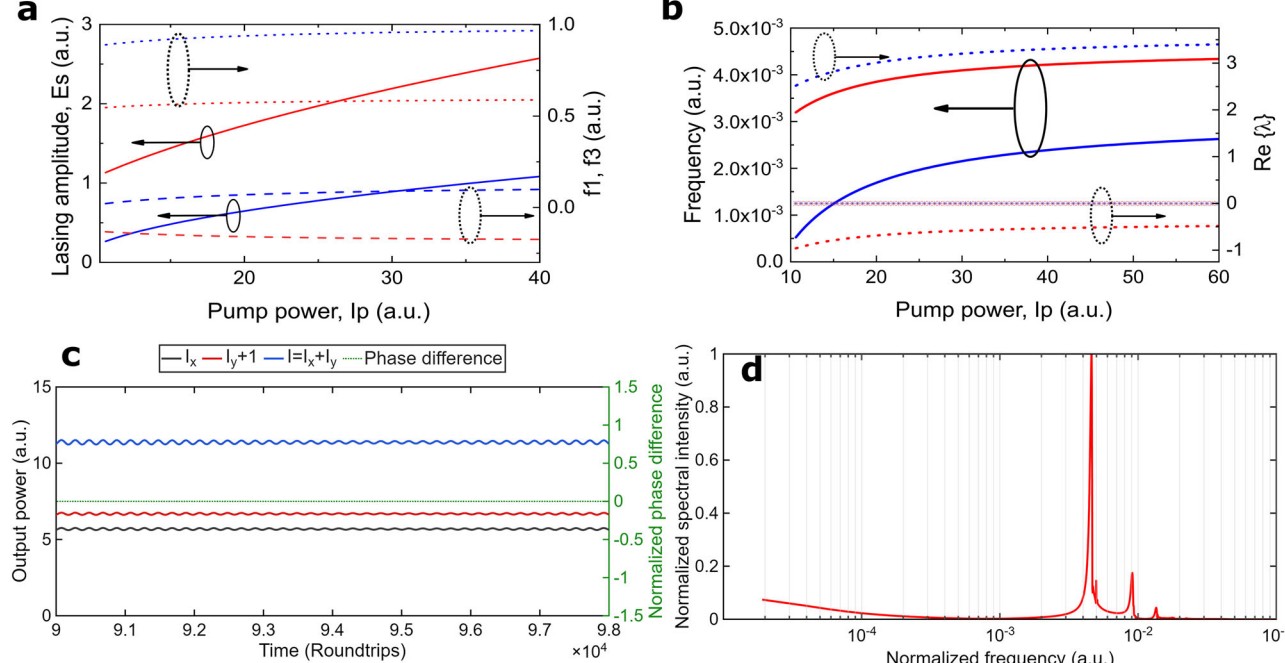

**Fig. 6 | Numerical simulation results.** Parameters (**a–d**): $\alpha_1 = 10.131$, $\varepsilon = 0.6 \cdot 10^{-5}$; $\alpha_2 = 2.3$, $\Delta = 0.015$, $\chi = 2.3$, $\gamma = 2 \cdot 10^{-6}$, $\delta = 1$; (**c**, **d**): $I_p = 35$: **a** Steady-state values of the lasing field as a function of the pump power $I_p$: $\Delta\varphi = 0$ (red colored lines), $\Delta\varphi = \pi$ (blue colored lines), $E_s$ (solid lines), $f_{1s}$ (dotted lines), $f_{3s}$ (dashed lines); **b** Imaginary and Real parts of the eigenvalues from Branch II as a function of the pump power $I_P$, $\Delta\varphi = 0$ (red line), $\Delta\varphi = \pi$ (blue line), $\lambda_3 = R_2$, (dashed lines), $\lambda_{4,5} = R_3 \pm i\Omega_{3,4}$; $\Omega_{3,4}$ (solid lines); $R_3$ (dotted line); $R_2 < 0$ ($\Delta\varphi = 0$, × line; $\Delta\varphi = \pi$, + line); **c** Polarization dynamics obtained by numerical solution of Eqs. (6–10) where $I_x$ (black) and $I_y$ (red), total power $I = I_x + I_y$ (blue), and the phase difference $\Delta\varphi$ between orthogonal x- and y-SOPs, normalized to $\pi$ (green, Secondary Y-axis); **d** Fast Fourier transform spectrum of the y-SOPs' power dynamics with frequencies normalized to the fundamental frequency.

presented[36,37]:

$$A = \ln(T) = \begin{bmatrix} A_{11} & A_{12} \\ A_{21} & A_{22} \end{bmatrix},$$

$$T = T_{POC1} \cdot T_{POL} \cdot T_{POC2}, \quad (i, j = 1, 2)$$

$$T_{POC1(2)} = \begin{bmatrix} \exp\left(\frac{i\varphi_{1(2)}}{2}\right)\cos(\xi_{1(2)}) & \exp\left(\frac{i\varphi_{1(2)}}{2}\right)\sin(\xi_{1(2)}) \\ -\exp\left(-\frac{i\varphi_{1(2)}}{2}\right)\sin(\xi_{1(2)}) & \exp\left(-\frac{i\varphi_{1(2)}}{2}\right)\cos(\xi_{1(2)}) \end{bmatrix},$$

$$T_{POL} = \begin{bmatrix} \sin(\theta)^2 \cdot \sqrt{1 - p^2} + p \cdot \cos(\theta)^2 & (p - \sqrt{1 - p^2}) \cdot \sin(\theta)\cos(\theta) \\ (p - \sqrt{1 - p^2}) \cdot \sin(\theta)\cos(\theta) & \cos(\theta)^2 \cdot \sqrt{1 - p^2} + p \cdot \sin(\theta)^2 \end{bmatrix}.$$

$$(11)$$

$T_{POC1(2)}$ and $T_{POL}$ are Jones matrices describing SOP transformation by POC1, POC2 and polarizer, where $\xi_{1(2)}$ is the angle of rotation of the vertical birefringent axis and $\varphi_{1(2)}$ is the phase shift between the wave components in the two orthogonal birefringent axes, the angle between the axis of the polarizer and the y axis is $\theta$. Matrix $A$ in Eq.11 accounts for the NPR of the lasing field. The functions $f_1$, $f_2$, $f_3$ are related to the angular distribution of the excited ions $n(\theta)$ expanded into a Fourier series as follows[26,38]:

$$n(\theta) = \frac{n_0}{2} + \sum_{k=1}^{\infty} n_{1k}\cos(k\theta) + \sum_{k=1}^{\infty} n_{2k}\sin(k\theta),$$

$$f_1 = \left(\chi\frac{n_0}{2} - 1\right), f_2 = \chi\frac{n_{12}}{2}, f_3 = \chi\frac{n_{22}}{2}.$$

$$(12)$$

Equations (6–10) have been derived by using the approximation that the dipole moments of the absorption and emission transitions for Er-doped silica are oriented randomly at angles $\theta$ in the plane orthogonal to the direction of light propagation[26,38]. In contrast to the more general assumption of the 3D distribution of the dipoles' orientations[39,40], an approximation (12) allows getting the finite dimension system presented by Eq. (6) where only $n_0$, $n_{12}$ and $n_{21}$ contribute to the dynamics of the lasing field.

We started our study with the case when the polarizer is oriented at angle $\theta = -\pi/4$, polarization extinction ratio $p = 0.99$, and there is no birefringence in the cavity, e.g., $\varphi_{1(2)} = \xi_{1(2)} = 0$ and $A_{11} = A_{22} = -1.1543$, $A_{12} = A_{21} \approx 1.1543$. For the case when there is no CW signal in the cavity ($a = 0$), we substitute $u = |u|\exp(i \cdot \varphi_x)$ and $v = |v|\exp(i \cdot \varphi_y)$ into Eq.6, and find the steady state solutions $|u| = |v| = E_{s0}$, $(E_{s\pi})$, $\Delta\varphi = \varphi_y - \varphi_x = 0, (\pi)$, $f_{1s} \neq 0$, $f_{2s} = 0$, $f_{3s} \neq 0$ as shown in Fig.6a, b As follows from Fig. 6a, the presence of two polarization controllers and polarizer in the cavity result in non-equal lasing field amplitudes for two orthogonal states of polarization with $\Delta\varphi = 0, \pi$.

To specify conditions for the NPR-based oscillations in the case of absence of CW laser radiation in the cavity, we linearize the Eqs.6–9 in the vicinity of the steady-state solutions mentioned above and find eigenvalues split in two branches I and II as follows (details are found in Supplementary Information Fig. S2).

The results for $\lambda_{4,5}$ in the case of $\Delta\varphi = 0, (\pi)$ and $\lambda_3$ ($\Delta\varphi = \pi$) are shown in Fig. 6b. As a result of non-equal amplitudes of the lasing fields for the orthogonal states with $\Delta\varphi = 0, \pi$ (Fig. 6a), the frequencies of oscillations nearby these states will be non-equal as well (Fig. 6b). Given the superposition of in-phase ($\Delta\varphi = 0$) and anti-phase oscillations ($\Delta\varphi = \pi$), the numerical solution of Eq.6 shows that the total output power will be oscillating as well (Fig. 6c). This modeling framework enables a comprehensive understanding of the vector dynamics governing VSHE and confirms the feasibility of entrainment and synchronization under realistic experimental conditions.

## Reporting summary

Further information on research design is available in the Nature Portfolio Reporting Summary linked to this article.

## Availability of Data and Materials

The datasets used and/or analyzed during the current study available from the corresponding author on the request.

## Code availability

The code that supports the plots within this paper is described in the Methods and Supplementary Information and is available from the corresponding author upon request.

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

## Acknowledgements

This work was supported by the UK Engineering and Physical Sciences Research Council (EPSRC) [grant number EP/W002868/1], Leverhulme Trust grant HARVEST [RPG-2023-073], National Natural Science Foundation of China [62135007], National Natural Science Foundation of China [61975107] and Natural Science Foundation of Shanghai Municipality [24ZR1422000]"111" project [D20031].

## Author contributions

D.S. and S.S. planned the project. D.S. carried out the experiment with help from S.S. D.S. and S.S. performed the data acquisition and edited the experimental figures. S.S. provided the theoretical support with help from H.K. and D.S. D.S., S.S., and C.M. wrote the manuscript with discussions with H.K., F.W., and Q.H.

## Competing interests

The authors declare no competing interests.
