## [Transparent Peer Review file · Communications Physics]

Nonlinear synchronization through vector subharmonic entrainment

Corresponding Author: Dr Dmitrii Stoliarov

Version 0:

Reviewer comments:

Reviewer #1

(Remarks to the Author)

The authors demonstrate a new type of subharmonic entrainment in the context of nonlinear synchronization. They unveil the vectorial nature of subharmonic entrainment, which has received little attention previously. It is unexpected that the additional polarization dimension of injection locking can control the intracavity dynamics of the laser, going beyond conventional intensity manipulation. The authors have also successfully developed a theoretical that quantitatively explains their observations. Overall, the submission is structurally sound and suitable for publication.

My comments are provided below to help the authors further improve the manuscript for clarity:

Although the authors discuss polarization synchronization, it appears to resemble a well-controlled difference between orthogonal polarization states. Can this type of synchronization be interpreted as a form of π -matching, which is typically essential for various nonlinear phenomena?

In a typical injection-locking system, the detuning of the injected signal and its intensity are critical parameters. The authors claim that the polarization dimension serves as a probe for intracavity dynamics, which is indeed interesting. However, would this nonlinear polarization coupling enhance the detuning-based intracavity control, or is it entirely independent of resonance detuning?

With an evolving polarization state of the injected beam, the laser oscillates in a well-controlled manner. However, the physical mechanism and quantitative description remain somewhat unclear. How can an external signal control slow pulsations that typically occur on the timescale of ion relaxation? Could resonance-facilitated gain depletion play a role?

The authors analyze power fluctuations of the polarization state using Fast Fourier Transform (FFT). The resulting power spectral density plotted against normalized frequency is used extensively throughout the manuscript. While FFT analysis is indeed valuable and rarely employed in similar studies, its specific relevance here requires further clarification. The differences observed in the FFT under controlled injection conditions also deserve more attention for a clear interpretation.

The coherence and quality of the figures need further improvement.

Reviewer #2

(Remarks to the Author)

Please see the attached file.

Reviewer #3

(Remarks to the Author)

The authors present their experimental work on subharmonic entrainment in a fiber laser. This work, while new, is not particularly exciting, poorly supported by the data and discussion, and more importantly, very badly presented. Overall, I

cannot recommend this work to be published in Communications Physics.

In more details:

- The authors keep on using the word "mode-locked laser", where they never show an actual mode-locked operation, only pseudo-CW and Q-switched mode-locked are shown. The following sentence in the caption of Fig 1 is particularly nonsensical: "the CW regime of the NPR mode-locked Er-doped fiber laser": If it's CW, then it's not mode-locked. The paper is littered with such approximations and inaccuracies that make it extremely difficult to read.
- The authors begin their discussion with an analogy with coupled spring. This could be very interesting if the authors provided an explanation of what is analogous to what: what is oscillating in their case? What is the coupling mechanism? How do you measure the oscillations, and how do you know when subharmonic entrainment is achieved? How do you justify its vectorial nature (and how is the analogy with springs and pendulum vectorial)?
- "This regime exhibits two-dimensional high-order partial synchronization across both temporal and polarization domains." Which quantity can be synchronized, how?
- "We seeded an optical power of 15 dBm at 1.67 kHz" This sentence does not have any meaning in physics, please elaborate, specifically on what is oscillating at 1.67 kHz, what kind of light has a power of 15 dBm.
- Where is the polarimeter located? It is not visible either in Fig. 1b (nor in the supplementary). The sentence that introduces it is also not grammatically correct.
- The CW operation is not qualified enough: the spectrum of fig 1c is not resolved enough to justify that this is CW. It even looks (qualitatively) that it's ~0.1nm wide, compatible with a few 10s of ps pulses, or pseudo random fluctuations.
- The VSHE is not clearly shown in Fig. 2 or Fig. 3, the authors should better describe the results they obtained and explain what they are looking for.
- Details on the mode-locking operation, on what is included in the , on the approximation made, ... should be detailed.

Less important issues:

- Equations are not properly aligned (1, 4, especially)
- SOP is not defined early enough.

Reviewer #4

(Remarks to the Author)

The manuscript reports the first theoretical and experimental observation of vector subharmonic entrainment (VSHE) in an ultrafast fiber laser, combining polarization-resolved measurements and an extended vector . The experimental evidence is clear, and the theoretical is detailed and consistent with observations. The paper is well organized, and the figures are informative. The study extends the concept of subharmonic entrainment to the vector regime and provides both physical insight and potential applications in photonic control and polarization-encoded ultrafast systems.

However, several points could be strengthened before publication:

1. The novelty should be more explicitly discussed in the Introduction.
2. The physical interpretation of the polarization -slip dynamics deserves clearer explanation.
3. The theoretical section is very long; key equations could be summarized in the main text and the full derivations moved to Supplementary Materials.
4. Minor language polishing is needed to improve flow and reduce redundancy.

Version 1:

Reviewer comments:

Reviewer #1

(Remarks to the Author)

I think the author has carefully addressed my concerns on the article, and the current version is acceptable for publication.

Reviewer #2

(Remarks to the Author)

The authors have done extensive revision which address most of reviewers comments. In my opinion, the revised paper can be published.

Reviewer #4

(Remarks to the Author)

The authors have significantly improved the manuscript in response to the previous comments. The presentation is now much clearer, and most of the major conceptual and terminological issues have been adequately addressed. In particular, the distinction between locking and entrainment is clearer, and the role of polarization dynamics in

subharmonic entrainment is better motivated both experimentally and theoretically. The revisions to the figures and captions also improve readability and consistency.

Overall, the manuscript is technically sound and much improved. I believe the remaining issues are minor and can be addressed with small textual clarifications.

Authors' Response to Reviews of

Nonlinear synchronization through vector subharmonic entrainment

Dmitrii Stoliarov^{1*}, Sergey Sergeyev¹, Hani Khashi¹, Fan Wu², Qianqian Huang², Chengbo Mou^{2*}

^{1*}Aston Institute of Photonics Technologies, Aston University, Birmingham, B4 7ET, UK.

²Key Laboratory of Specialty Fiber Optics and Optical Access Networks, Joint International Research Laboratory of Specialty Fiber Optics and Advanced Communication, Shanghai University, Shanghai, 200444, China.

RC: Reviewers' Comment, AR: Authors' Response, ___: Revised Manuscript Text

Dear Reviewers,

We thank you all for your careful review of our manuscript. We are grateful for the many constructive and encouraging comments - reviewers explicitly assessed the work positively - which helped us substantially improve the clarity, completeness, and impact of the paper. Across the reports, reviewers raised overlapping points all of which we have addressed of these shared issues - along with all additional, reviewer-specific concerns - by restructuring the main text, clarifying the theoretical framework, expanding our analyses, and revising the figures. The Supplementary Information has also been expanded to address questions raised by the reviewers. We believe these changes materially improve readability and highlight the novelty and generality of our findings.

How to read our point-by-point responses:

In the following pages, we respond point-to-point to all comments. Because several concerns were shared, we sometimes use the same revised text section.

Reviewer comments are labelled as RC#

Our replies are numbered as AR# (Author Response).

We have included verbatim excerpts from the manuscript and have underlined the text that has been revised or newly added. In the revised manuscript, all amendments are highlighted in blue for easy identification and underlined.

We appreciate the reviewers' thoughtful feedback and believe the revisions substantially strengthen the manuscript. We hope the improved clarity, expanded analyses, and new results address all concerns satisfactorily.

Reviewer 1.

RC1.1: Although the authors discuss polarization synchronization, it appears to resemble a well-controlled phase difference between orthogonal polarization states. Can this type of phase synchronization be interpreted as a form of phase-matching, which is typically essential for various nonlinear phenomena?

AR1.1: The phase locking between the orthogonally polarized state of polarization is shown in Figs. 1d and 6c. As follows from Figs. 2-5, there is no the phase locking between the orthogonally polarized states, it is so-called phase entrainment, e.g. phase difference oscillations. The phase difference is oscillating whereas the coupling between high-frequency and low-frequency oscillations results in pulse bunching (Q-switch mode locking).

RC1.2: In a typical injection-locking system, the detuning of the injected signal and its intensity are critical parameters. The authors claim that the polarization dimension serves as a probe for intracavity dynamics, which is indeed interesting. However, would this nonlinear polarization coupling enhance the detuning-based intracavity control, or is it entirely independent of resonance detuning?

AR1.2: We updated Figs 4 and 5 to clarify our approach. The same as for typical injection-locking, detuning in terms of increased frequency of the injected signal (Fig. 4 c, d) and amplitude (Fig. 5) are critical parameters defining the output waveform. But, unlike typical injection-locking, the changing polarization dynamics of the injection signal enhances the detuning-based control as shown in Figs. 5 c, d.

RC1.3: With an evolving polarization state of the injected beam, the laser oscillates in a well-controlled manner. However, the physical mechanism and quantitative description remain somewhat unclear. How can an external signal control slow pulsations that typically occur on the timescale of ion relaxation? Could resonance-facilitated gain depletion play a role?

AR1.3: To clarify mechanism of the control we added a paragraph in the Discussion section:

The new distinct feature of vector SHE is in dependence of the synchronization regime on the polarization waveform of the injected signal, as shown in Figs. 4 and 5. The shape of the SOP of the injected signal with the rotating state of polarization enables control of the dwelling time near each orthogonal SOP, resulting in different types of switching between the orthogonal SOPs, taking the form of different QSML dynamics.

RC1.4: The authors analyze power fluctuations of the polarization state using Fast Fourier Transform (FFT). The resulting power spectral density plotted against normalized frequency is used extensively throughout the manuscript. While FFT analysis is indeed valuable and rarely employed in similar studies, its specific relevance here requires further clarification. The differences observed in the FFT under controlled

injection conditions also deserve more attention for a clear interpretation.

AR1.4: To clarify the FFT analysis, we added paragraph in the Discussion section:

The FFT of the dynamic waveforms demonstrate the increased number of the sidebands around the high-frequency component with increased amplitude and varying dynamic SOP of the injection signal, which indicates enhanced coupling between the high- and low-frequency oscillations.

RC1.5: The coherence and quality of the figures need further improvement.

AR1.5: To improve the coherence and quality of the figures, all of them (Fig 1-6, Fig.1S) have been revised, and Fig. 2S was added to improve the justification of the similarity between the experimental and theoretical results.

Reviewer 2

RC2.1: Synchronization can be well described by the Adler equation. For discussion, is there any nonlinear version of the Adler equation that could possibly interpret the phenomenon observed in this paper?

AR2.1: To clarify the synchronization scenarios based on the Adler equation, we added the following paragraph in Introduction section:

The spring-coupled simple pendulum system Fig. 1a provides a classical analogue to the dynamics of orthogonally polarized modes in mode-locked lasers, where each pendulum represents an oscillator associated with one of the state of polarization (SOP) [8, 9, 26, 28, 30]. The synchronization regimes of the coupled oscillators can be explained by the general Adler equation, describing the evolution of the phase difference $\Delta\phi$ between two coupled oscillators [8, 9, 26, 28, 30]:

$$\frac{d\Delta\phi}{dt} = \Delta\Omega + K \cdot \sin(\Delta\phi), \quad (1)$$

where $\Delta\Omega$ is the frequency difference, and K is the coupling coefficient. For polarization dynamics of the mode-locked lasers, the frequency difference is the function of the linear and circular birefringence, and the coupling coefficient is a function of the output powers of the orthogonal SOPs [26, 28, 30]. As follows from the Eq. (1), the phase-locked synchronization ($d\Delta\phi/dt = 0$) exists when $|\Delta\Omega| < |K|$ that corresponds to the continuous-wave (CW) mode-locking [26, 28, 30]. Unlike the phase-locking, condition $|\Delta\Omega| > |K|$ means the phase entrainment, e. g. phase difference oscillations [8, 26, 28, 30]. In contrast to the classical Adler equation with constant coefficients, the generalized Adler equation derived for the polarization laser dynamics comprises the time-dependent

coefficients [26, 28, 30]. As a result, condition of the phase entrainment has to be generalized [28]. Though differences between mechanical model and vector mode-locked laser, phase-difference dynamics in mode-locked fibre lasers can be treated in terms of coupling-induced transitions [26, 28, 30].

RC2.2: For injection locking, typical frequency detuning and intensity require delicate control. However, this does not seem to be the case in this submission. Can polarization itself dominate the injection locking?

AR2.2: We updated Figs. 4 and 5 to demonstrate that the detuning, amplitude and the state of polarization of the injected signal plays a crucial role. The same as for typical injection-locking, detuning in terms of increased frequency of the injected signal (Fig. 4 c, d) and amplitude (Fig. 5) are critical parameters defining the output waveform. But, unlike typical injection-locking, the changing polarization dynamics of the injection signal enhances the detuning-based control as shown in Figs. 5 c, d

RC2.3: The authors mentioned that the proposed approach can be used for laser intensity control because they use the external probe to mode-lock and Q-switch a laser. Would this be practically useful? Any other laser operation regime might be obtained?

AR2.3:As follows from the updated Figs. 4 and 5, control of the amplitude, frequency and the state of polarization of the injected signal enables manipulation of the output lasing waveform. During the experimental work, many regimes have been achieved, including mode locking (see Figs. 1S b and c in supplementary materials), harmonic mode locking, continuous-wave and Q-switching. However, these regimes are outside the scope of our proposed concept and were not the subject of our study.

RC2.4: I suggest improving the figure quality, for example, Fig.2 should be improved for a nice piece of artwork.

AR2.4: Fig. 2 has been updated, as requested.

Reviewer 3

RC3.1: The authors keep on using the word "mode-locked laser", where they never show an actual mode-locked operation, only pseudo-CW and Q-switched mode-locked are shown. The following sentence in the caption of Fig 1 is particularly nonsensical: "the CW regime of the NPR mode-locked Er-doped fiber laser": If it's CW, then it's not mode-locked. The paper is littered with such approximations and inaccuracies that make it extremely difficult to read.

AR3.1:The optical spectra and oscilloscope traces of the laser setup operating in the mode-locked regime were included in the supplementary materials (see Figs. 1S b and c). Caption of Fig 1 was revised to address the reviewer comment:

Fig. 1 Principle and schematic of subharmonic entrainment induced nonlinear synchronization: (a) Simple spring-coupled oscillator model: two pendulums of different mass and unequal lengths l_1 and l_2 are coherently coupled via a spring of finite stiffness. The angular displacements φ_1 and φ_2 describe the phase evolution of each oscillator, which possess distinct natural frequencies. (b) Schematic setup of the fiber laser. EDF: erbium-doped fiber; 980 nm pump laser diode; 1550 nm continuous wave laser: CW laser 1550 nm; FPC: manual fiber polarization controller; EPC: electronically driven polarization controller; ISO polarization-sensitive optical isolator; WDM: wavelength-division multiplexer; PS: polarization scrambler; OC: 70/30 optical coupler, OSA: optical spectrum analyzer (c) Optical spectrum of the CW regime of Er-doped fiber laser. (d) Polarization dynamics of Er-doped fiber laser in terms of oscillations of the output powers of the individual polarization components I_x (black) and I_y (red), total power $I = I_x + I_y$ (blue), and the phase difference $\Delta\varphi$ between orthogonal x- and y-SOPs normalized to π (green). (e) Fast Fourier transform spectrum of the x- and y-SOPs' power dynamics with frequencies normalized to the fundamental frequency $f_0 = 16.67$ MHz. Parameters: pump power $I_p = 72$ mW, averaging time for the polarimeter trace $T_{pol} = 320$ ns.

Along with the text in the Result section:

In the absence of the injected signal, the electronically driven polarization controller (EPC) was tuned to establish the CW regime, as depicted in the optical spectrum shown in Fig. 1(c). As illustrated, the optical spectrum exhibits a localized, narrowband spike-like structure centred around 1562 nm. The polarization resolved temporal behaviours of pulses, shown in Fig. 1 d, appear as a close to CW low-amplitude oscillations in the output powers of the orthogonal SOPs, I_x and I_y , the total power $I = I_x + I_y$, and the phase difference $\Delta\varphi$. A fast Fourier transform (FFT) reveals that the oscillation frequency is $f_{cw} = 2 \times 10^{-3} \cdot f_0$ for the x- and y-SOPs, as shown in Fig. 1 e.

The additional text has been added to the supplementary material as follows:

The stable mode-locked regime was achieved by adjusting the polarization controllers in the absence of CWL injection. Figures 1b and c demonstrate the optical spectra and oscilloscope trace of the stable single-pulse mode-locked regime with a fundamental pulse repetition rate of 16.67 MHz.

RC3.2: The authors begin their discussion with an analogy with coupled spring. This could be very interesting if the authors provided an explanation of what is analogous to what: what is oscillating in their case? What is the coupling mechanism? How do you measure the oscillations, and how do you know when subharmonic entrainment is achieved? How do you justify its vectorial nature (and how is the analogy with springs and pendulum vectorial)? How do you measure the oscillations, and how do you know when subharmonic entrainment is achieved? How do you justify its vectorial nature (and how is the analogy with springs and pendulum vectorial)?

AR3.2: To clarify the analogy of the synchronization scenarios between coupled oscillators and the orthogonally polarized states of polarization in mode-locked laser, we added the following paragraph in Introduction section:

The spring-coupled simple pendulum system Fig. 1a provides a classical analogue to the dynamics of orthogonally polarized modes in mode-locked lasers, where each pendulum represents an oscillator associated with one of the state of polarization (SOP) [8, 9, 26, 28, 30]. The synchronization regimes of the coupled oscillators can be explained by the general Adler equation, describing the evolution of the phase difference $\Delta\phi$ between two coupled oscillators [8, 9, 26, 28, 30]:

$$\frac{d\Delta\phi}{dt} = \Delta\Omega + K \cdot \sin(\Delta\phi), \quad (1)$$

where $\Delta\Omega$ is the frequency difference, and K is the coupling coefficient. For polarization dynamics of the mode-locked lasers, the frequency difference is the function of the linear and circular birefringence, and the coupling coefficient is a function of the output powers of the orthogonal SOPs [26, 28, 30]. As follows from the Eq. (1), the phase-locked synchronization ($d\Delta\phi/dt = 0$) exists when $|\Delta\Omega| < |K|$ that corresponds to the continuous-wave (CW) mode-locking [26, 28, 30]. Unlike the phase-locking, condition $|\Delta\Omega| > |K|$ means the phase entrainment, e. g. phase difference oscillations [8, 26, 28, 30]. In contrast to the classical Adler equation with constant coefficients, the generalized Adler equation derived for the polarization laser dynamics comprises the time-dependent coefficients [26, 28, 30]. As a result, condition of the phase entrainment has to be generalized [28]. Though differences between mechanical model and vector mode-locked laser, phase-difference dynamics in mode-locked fibre lasers can be treated in terms of coupling-induced transitions [26, 28, 30].

We updated the figures 4 and 5 to justify that amplitude, the frequency of modulation and the state of polarization are the crucial parameters for the vector subharmonic entrainment.

Additionally, we updated the paragraph in the Discussion section to justify the analogy and distinct features of vector subharmonic entrainment versus coupled oscillators with external forcing:

According to the synchronization theory for subharmonic entrainment in our case, the ratio of frequencies can slightly deviate from an integer, which results in the synchronization in the form of phase difference entrainment, e.g., the phase difference oscillations shown in [9, 25, 36] and Figs. 2, 3, and 4. With an increased amplitude of the injected signal (see Fig. 4), in line with the synchronization theory [9, 25, 36], we observe phase- and frequency locking towards high-power oscillations. To obtain experimentally such dynamics, it is necessary to adjust the power and wavelength of the internal low-frequency oscillations, which is rather cumbersome for discussion here and will be published elsewhere. The new distinct feature of vector SHE is in dependence of the synchronization regime on the polarization waveform of the injected

signal, as shown in Figs. 4 and 5. The shape of the SOP of the injected signal with the rotating state of polarization enables control of the dwelling time near each orthogonal SOP, resulting in different types of switching between the orthogonal SOPs, taking the form of different QSML dynamics.

RC3.3: "This regime exhibits two-dimensional high-order partial synchronization across both temporal and polarization domains." Which quantity can be synchronized, how?

AR3.3: In Figs. 1d, 2c, 3b, 4a,c; 5a,c; 6c we presented the phase difference dynamics for the orthogonally polarized modes. As follows from the figures, the synchronization scenarios between the state of polarization take the form of phase locking in Figs. 1 d and 6 c and the phase entrainment shown in 2c, 3b, 4a,c; 5a,c.

RC3.4: "We seeded an optical power of 15 dBm at 1.67 kHz" This sentence does not have any meaning in physics, please elaborate, specifically on what is oscillating at 1.67 kHz, what kind of light has a power of 15 dBm.

AR3.4: We revised the sentence to address the reviewer's comment:

We seeded an optical signal with power of 15 dBm and modulation frequency of 1.67 kHz through a vacant input port of a 30/70 output optical coupler.

RC3.5: Where is the polarimeter located? It is not visible either in Fig. 1b (nor in the supplementary). The sentence that introduces it is also not grammatically correct.

AR3.5: The fig 1 b has been revised to address the question of the reviewer. The sentence was edited as follows to address the reviewer's comment:

A fast polarimeter PM1000-XL-FA-N20 D (Novoptel) with a sampling frequency of 100 MS/s was used to measure the evolution of the Stokes parameters \$S_0\$, \$S_1\$, \$S_2\$, and \$S_3\$. The power for the orthogonal x- and y- polarization components (\$I_x\$, \$I_y\$ ), the total power \$I\$, and the phase difference \$\Delta\phi\$ was recalculated from the measured Stokes parameters as follows:

RC3.6:The CW operation is not qualified enough: the spectrum of fig 1c is not resolved enough to justify that this is CW. It even looks (qualitatively) that it's ~0.1nm wide, compatible with a few 10s of ps pulses, or pseudo random fluctuations.

AR3.6:We updated text and Fig in Results section as follows to clarify:

In the absence of the injected signal, the electronically driven polarization controller (EPC) was tuned to establish the CW regime, as depicted in the optical spectrum shown in Fig. 1(c). As illustrated, the optical spectrum exhibits a localized, narrowband spike-like structure centred around 1562 nm. The polarization resolved temporal behaviours of pulses, shown in Fig. 1 d, appear as a close to CW low-amplitude oscillations in the output powers of the orthogonal SOPs, \$I_x\$ and \$I_y\$, the total power \$I = I_x + I_y\$, and the phase difference \$\Delta\phi\$. A fast Fourier transform (FFT) reveals that the oscillation frequency is \$f_{cw} = 2 \times 10^{-3} \cdot f_0\$ for the x- and y-SOPs, as shown in Fig. 1 e.

Fig. 1c was measured at the same resolution as all the other spectra in the work. The following text added to the Methods section:

All optical spectra were measured at the same wavelength resolution of 0.1 nm.

RC3.7:The VSHE is not clearly shown in Fig. 2 or Fig. 3, the authors should better describe the results they obtained and explain what they are looking for.

AR3.7:We describe the experimental results for VSHE in the context of fast- and slow-frequency signal interaction resulting in the appearance of the side modes around the high frequency oscillation (section Results):

The presence of sidebands in the high-frequency oscillations is a result of overlapping with the low-frequency components, thereby enabling partial synchronization through the SHE mechanism.

The results of theoretical analysis shown in Figs. 4-6 help to clarify the mechanism of VSHE in more details (section Discussions):

The observed SHE represents a distinct synchronization phenomenon where, as shown in [9, 25], the synchronization behaviour depends on both the injected signal amplitude (or coupling coefficient) and the detuning between the injected modulated frequency and the internal oscillation frequency of the system. According to the synchronization theory for subharmonic entrainment in our case, the ratio of frequencies can slightly deviate from an integer, which results in the synchronization in the form of phase difference entrainment, e.g., the phase difference oscillations shown in [9, 25, 36] and Figs. 2, 3, and 4. With an increased amplitude of the injected signal (see Fig. 4), in line with the synchronization theory [9, 25, 36], we observe phase- and frequency locking towards high-power oscillations. To obtain experimentally such dynamics, it is necessary to adjust the power and wavelength of the internal low-frequency oscillations, which is rather cumbersome for discussion here and will be published elsewhere. The new distinct feature of vector SHE is in dependence of the synchronization regime on the polarization waveform of the injected signal, as shown in Figs. 4 and 5. The shape of the SOP of the injected signal with the rotating state of polarization enables control of the dwelling time near each orthogonal SOP, resulting in different types of switching between the orthogonal SOPs, taking the form of different QSML dynamics.

RC3.8: Details on the mode-locking operation, on what is included in the model, on the approximation made, ... should be detailed.

AR3.8: We addressed the reviewer's comment in the main text (section Results):

To overcome the drawbacks of previously used model [36] and so reveal mechanism of the mode-locking dynamics driven by the SHE between low-frequency oscillations, we updated a vector model of an Er-doped mode-locked fibre laser recently developed by Sergeyev and co-workers (details are found in [26–31]). The model describes the evolution of the SOP of the lasing averaged over the pulse width driven by in-cavity CW signal with periodically evolving orthogonal states of polarization:

RC3.9: Less important issues:

- Equations are not properly aligned (1, 4, especially)
- SOP is not defined early enough.

AR3.9: The reviewer's comments have been addressed in the main text.

Reviewer 4

RC4.1: The novelty should be more explicitly discussed in the Introduction.

AR4.1: We updated the introduction section as follows to address the reviewer's comment:

In this work, we report for the first time the experimental and theoretical demonstration of vector subharmonic entrainment (VSHE) in an ultrafast laser. We experimentally observe synchronization between a low-frequency, injected, polarization-modulated signal and internal polarization oscillations in an ultrafast laser. This phenomenon results in a double-timescale pulsation regime, also known as Q-switched mode-locked (QSML), characterized by a train of ultrashort pulses modulated by a slowly varying envelope. The observed synchronization scenarios were mapped by the phase difference dynamics and were adjusted by varying the injected signal modulation frequency, amplitude, and the state of polarization synchronization regime.

RC4.2: The physical interpretation of the polarization phase-slip dynamics deserves clearer explanation.

AR4.2: We added the following paragraph to the section Discussion to address the reviewer's comment:

The new distinct feature of vector SHE is in dependence of the synchronization regime on the polarization waveform of the injected signal, as shown in Figs. 4 and 5. The shape of the SOP of the injected signal with the rotating state of polarization enables control of the dwelling time near each orthogonal SOP, resulting in different types of switching between the orthogonal SOPs, taking the form of different QSML dynamics.

RC4.3: The theoretical model section is very long; key equations could be summarized in the main text and the full derivations moved to Supplementary Materials.

AR4.3: We have revised the text and left only the key equations summarised in the Methods section. All derivations related to the steady-state stability analysis have been moved to the Supplementary Materials.

RC4.4: Minor language polishing is needed to improve flow and reduce redundancy.

AR4.4: The text was edited to improve the manuscript's style towards more clear physical interpretation of vector subharmonic entrainment. The revised text is highlighted in blue.

Authors' Response to Reviews of Nonlinear synchronization through vector subharmonic entrainment

Dmitrii Stoliarov^{1*}, Sergey Sergeyev¹, Hani Khashi¹, Fan Wu², Qianqian Huang², Chengbo Mou^{2*}

^{1*}Aston Institute of Photonics Technologies, Aston University, Birmingham, B4 7ET, UK.

²Key Laboratory of Specialty Fiber Optics and Optical Access Networks, Joint International Research Laboratory of Specialty Fiber Optics and Advanced Communication, Shanghai University, Shanghai, 200444, China.

Dear Reviewers,

We sincerely thank all reviewers for their careful evaluation of our manuscript and for their constructive feedback. We are pleased that Reviewer #1 and Reviewer #2 find that their concerns have been fully addressed and consider the revised manuscript suitable for publication. We also thank Reviewer #4 for recognizing the substantial improvements in clarity, terminology, and presentation, as well as for highlighting the clearer distinction between phase locking and phase entrainment and the improved discussion of polarization dynamics. We have carefully addressed the remaining minor points through additional clarifications in the text. We believe that these revisions have further strengthened the manuscript and we appreciate the reviewers' positive assessment.

REVIEWERS' COMMENTS

Reviewer 1.

I think the author has carefully addressed my concerns on the article, and the current version is acceptable for publication.

Reviewer 2.

The authors have done extensive revision which address most of reviewers comments. In my opinion, the revised paper can be published.

Reviewer 4.

The authors have significantly improved the manuscript in response to the previous comments. The presentation is now much clearer, and most of the major conceptual and terminological issues have been adequately addressed.

In particular, the distinction between phase locking and phase entrainment is clearer, and the role of polarization dynamics in subharmonic entrainment is better motivated both experimentally and theoretically. The revisions to the figures and captions also improve readability and consistency.

Overall, the manuscript is technically sound and much improved. I believe the remaining issues are minor and can be addressed with small textual clarifications.

Subharmonic entrainment is something correlated with synchronisation. In this paper, the authors show vector subharmonic entrainment through nonlinear synchronization. The authors utilised the vectorial feature of a weakly birefringent laser cavity to explore such phenomena. In particular, the authors involve nonlinear synchronisation where the mode-locked laser cavity is an intrinsically high-nonlinear system. The injected signal is similar to injection locking, but with a distinct difference that provides the capability to control the laser operation regimes through polarisation control. These experimental results have also been evidenced by the developed theoretical model. In general, I think this paper is suitable for publication upon several minor issues:

1. Synchronisation can be well described by the Adler equation. For discussion, is there any nonlinear version of the Adler equation that could possibly interpret the phenomenon observed in this paper?
2. For injection locking, typical frequency detuning and intensity require delicate control. However, this does not seem to be the case in this submission. Can polarization itself dominate the injection locking?
3. The authors mentioned that the proposed approach can be used for laser intensity control because they use the external probe to mode-lock and Q-switch a laser. Would this be practically useful? Any other laser operation regime might be obtained?
4. I suggest improving the figure quality, for example, Fig.2 should be improved for a nice piece of artwork.